# Long-term imaging of dorsal root ganglia in awake behaving mice

Chao Chen[1,5], Jinhui Zhang[2,5], Linlin Sun[3,5], Yiling Zhang[1], Wen-Biao Gan[4], Peifu Tang[1] & Guang Yang ⬤ [3]

The dorsal root ganglia (DRG) contain the somas of first-order sensory neurons critical for somatosensation. Due to technical difficulties, DRG neuronal activity in awake behaving animals remains unknown. Here, we develop a method for imaging DRG at cellular and subcellular resolution over weeks in awake mice. The method involves the installation of an intervertebral fusion mount to reduce spinal movement, and the implantation of a vertebral glass window without interfering animals' motor and sensory functions. In vivo two-photon calcium imaging shows that DRG neuronal activity is higher in awake than anesthetized animals. Immediately after plantar formalin injection, DRG neuronal activity increases substantially and this activity upsurge correlates with animals' phasic pain behavior. Repeated imaging of DRG over 5 weeks after formalin injection reveals persistent neuronal hyperactivity associated with ongoing pain. The method described here provides an important means for in vivo studies of DRG functions in sensory perception and disorders.

[1] Department of Orthopaedics, Peking 301 Hospital, Beijing 100853, China. [2] Department of Orthopaedics, the Affiliated Southeast Hospital of Xiamen University, Zhangzhou 175 Hospital, Zhangzhou 363000, China. [3] Department of Anesthesiology, Columbia University, New York 10032 NY, USA. [4] Department of Neuroscience and Physiology, Department of Anesthesiology, Skirball Institute, New York University School of Medicine, New York 10016 NY, USA. [5]These authors contributed equally: Chao Chen, Jinhui Zhang, Linlin Sun. Correspondence and requests for materials should be addressed to W.-B.G. (email: gan@saturn.med.nyu.edu) or to P.T. (email: tangpeifubeijing@163.com) or to G.Y. (email: gy2268@cumc.columbia.edu)

As the first-order neurons in the somatosensory system, sensory neurons in the dorsal root ganglion (DRG) detect peripheral stimuli and convey signals to the central nervous system (CNS)[1]. DRG neurons are pseudo-unipolar cells that are diverse in size[2], molecular constitution[3], electrophysiological properties[4], and myelination levels[5]. In the DRG, the somas of sensory neurons are tightly enwrapped by satellite glial cells[6]. Given the critical role of DRG in initiating somatosensation and pain, the functions of DRG cells have been extensively studied in cultured or isolated preparations[7–10] or in anesthetized animals[11,12]. However, the activity of DRG cells in awake behaving animals remains unclear due to the lack of suitable methods of measurement.

With recent advances of optical and genetic methods, in vivo imaging has been achieved in multiple cell types in a variety of tissues and organs of the living animals[13]. For example, synapses[14], neurons[15,16], and non-neuronal cells[17] in the mouse cortex can be repeatedly imaged via cranial windows in both health and disease. In the spinal cord, a laminectomy technique has been developed for in vivo imaging of spinal neuron activity[18], axonal degeneration and regeneration[19–22], microglial motility[19], as well as the dynamics of infiltrating immune cells and tumor growth[18,23–25]. Although DRG is in the vicinity of the spinal cord, in vivo imaging of DRG cells remains challenging for awake behaving mice. The established spinal imaging approaches are not suitable for imaging DRG because the laminectomy surgery removes the vertebrae holding the DRG, which made it difficult to manage motion artifacts during image collection. Previous studies have reported that following laminectomy, sensory neurons in the DRG could be imaged in anesthetized mice while their breathing is being suppressed[11]. However, this method restricts the duration of image collection to <6 h and prohibits more than one imaging session. Moreover, because neuronal activities under anesthesia are substantially different from that under awake states[26,27], the activity of sensory ganglion neurons during natural sensation remains elusive.

Spinal fusion is used in the treatment of patients with spinal instability[28,29]. In this study, we develop an intervertebral fusion method in mice, which minimizes spinal movement and allows chronic imaging of DRG neurons at cellular and subcellular resolution in awake behaving states. We demonstrate that DRG sensory neurons are more active in awake mice than in anesthetized mice. With formalin-induced inflammatory pain, canonical nociceptive neurons in the DRG exhibit phase-specific activity patterns, with the increase of activity positively correlating with the animals' spontaneous pain behavior. By longitudinally monitoring the activity of individual neurons in the DRG of awake mice up to 5 weeks, we demonstrate that hyperactivity of DRG sensory neurons persists in mice with chronic pain. The method developed in this study thus provides an important tool for in vivo studies of sensory ganglia-related disorders and neuropathies.

## Results

### Design of a vertebral window for imaging DRG in vivo.
In order to image DRG in the living mice, we custom designed a vertebral mount that could be firmly attached to the spinal column (Fig. 1a–c), while allowing access to the DRG through a 2-mm glass window (Fig. 1d). The vertebral mount was composed of two L-shaped stainless steel plates (one long, one short), one screw and matched washer that were registered with the screw hole on the short plate (Fig. 1b and Supplementary Figs. 1 and 2). The mounting plates consisted of prongs that could be inserted alongside the lateral aspects of the vertebral column (Fig. 1c), with the short and long plates aligned to the opposite

sides of the targeted DRG, respectively (Fig. 1e, f). Two plates were secured to the spinal column by screwing to each other (Fig. 1f). To prevent the spinal column from rotating so as to improve the image stability, the prongs of mounting plates were machined with anti-slipping surface (Supplementary Figs. 1 and 2, detail A). Mechanical drawings for custom parts are provided in Supplementary Figs. 1 and 2.

### Surgical procedures for implanting the vertebral window.
To implant the custom-designed vertebral window for in vivo imaging of DRG, the animal was deeply anesthetized and a skin incision was made on the back of the animal (see the "Methods" section for details). To create a window for the DRG located at the level of the fourth lumbar (L4) vertebra, muscles at the lateral aspects of L4 and L5 were retracted with an aluminum sheet (Fig. 1g). After the vertebral mount was inserted and secured to the spinal column, the articular processes surrounding the targeted DRG were trimmed to the same level as the DRG surface (Fig. 1h and Supplementary Fig. 3). A thin layer of silicone elastomer was subsequently applied to the surface of DRG and sealed with a 2-mm cover glass (Fig. 1d and h). A combination of cyanoacrylate and dental acrylic were used to attach the coverslip to the vertebral mount (Fig. 1i). Following a successful surgery, DRG can be clearly visualized through the glass window (Fig. 1i). The animals were then returned to their home cages for recovery from surgery. No signs of lordosis or kyphosis were observed over weeks after the vertebral mount and glass window implantation (Fig. 1j).

### Implantation has no effects on motor and sensory functions.
To assess the potential impact of vertebral window implantation on the animals' motility, we first tested the animals' gait patterns during voluntary running on a plexiglass enclosed track[30] (Fig. 2a). The patterns of footprints were analyzed by measuring the stride length in non-operated or sham-operated controls and mice with implantation. Sham-operated mice underwent skin incision and muscle/ligament detachment, but did not have the vertebral window implanted. We found that there was no significant difference in the hindlimb stride length on ipsilateral and contralateral sides at various time points after surgery between mice with and without implantation (Fig. 2a). We also assessed cumulative time of immobility, grooming, rearing[31], and speed distributions[32] during a 5-min open-field test. The grooming time in mice with implanted window was slightly higher than that in sham mice 1 day after surgery, but recovered to the similar level as in sham mice 4 days after surgery (Fig. 2b). No significant difference in immobility or rearing time was found between these two groups (Fig. 2b). Analysis of locomotor behavior revealed no differences in the speed distribution, average or top speed between sham and implanted mice 1–7 days after surgery (Fig. 2c–e). We also assessed the animals' mechanical sensation using von Frey filaments[33]. Upon pressure application to the plantar surface, mice with window implantation showed a paw withdrawal threshold that was comparable to that in non-operated or sham-operated mice 1–7 days after surgery (Fig. 2f). Collectively, these results demonstrate that vertebral window implantation has no substantial effects on the animals' locomotor and sensory functions.

To determine whether vertebral window implantation causes inflammation in the DRG, we performed histological analysis 1, 4, and 7 days after surgery using Cx3cr1-GFP mice expressing green fluorescent protein (GFP) in macrophages (Fig. 2g). We found that macrophage density was higher in the DRG of implanted mice as compared to sham-operated mice within 1 week after surgery (Fig. 2g, h). The density of macrophages peaked about

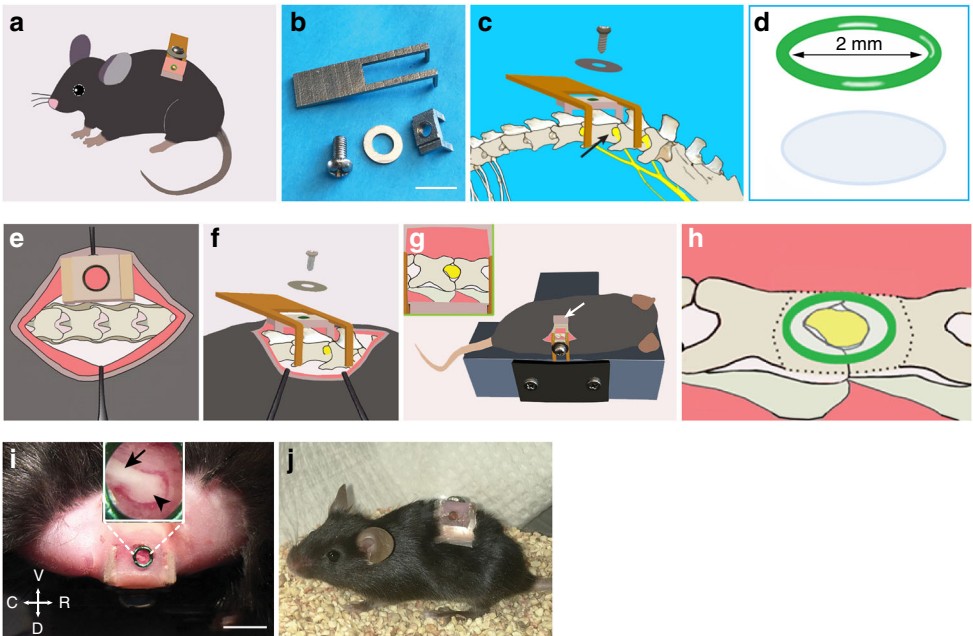

**Fig. 1** A vertebral window allowing DRG imaging in awake mice. **a** Schematic showing a mouse with an implanted DRG imaging window. **b** Components of the vertebral mount: two L-shaped mounting plates, one screw and one screw washer. Scale bar, 10 mm. **c** Schematic showing the assembly of the vertebral mount along L4 and L5 vertebrae, just above the iliac crest. L3–5 DRGs and spinal nerves were depicted in yellow. Arrow indicates L4 DRG. **d** Imaging window was made of a 2-mm diameter coverslip and aluminum ring (green). **e** Prongs of the short plate were aligned to the vertebral column on the opposite side of the targeted DRG. **f** Prongs of the long plate were positioned to the same side of the targeted DRG. Two plates were registered with a screw and a washer. **g** L4 DRG exposure. The mouse was positioned lying on its right side with the vertebral mount secured to a metal base. Arrow indicates the aluminum sheet used to retract lateral muscle. The inset shows the targeted DRG and the surrounding vertebral bone. **h** The articular processes surrounding the DRG was trimmed before the glass window (green circle) was placed on top of the DRG. Dotted line indicates the trimmed area. **i** Photograph showing the DRG window mounted in place with dental cement. The inset shows a zoom-in view of L4 DRG, ready for two-photon imaging. Arrow head points to DRG; arrow points to spinal nerve. C: caudal; R: rostral; D: dorsal; V: ventral. Scale bar, 5 mm. **j** Photograph showing a mouse with an implanted DRG window 14 days after the surgery

4 days after surgery in implanted mice. These results indicate that implantation surgery caused a mild inflammation in the DRG within the first week after the surgery.

**Long-term imaging of DRG neurons in awake mice.** Through the implanted vertebral window, we performed in vivo two-photon imaging of the DRG in *Thy1*-YFP mice that expressed yellow fluorescent protein (YFP) in a subset of sensory neurons. As shown in Fig. 3a, the somas and axons of sensory neurons in the DRG could be visualized through the glass window right after surgery and for at least 1 month after window implantation. To assess the image quality over time, we calculated the image contrast ratio by $(I_{max}-I_{min})/(I_{max} + I_{min})$ with $I_{max}$ and $I_{min}$ representing the highest and lowest fluorescence intensity. The image contrast ratios remained in the range between 0.8 and 0.95, with a slight decrease after 1 month (Fig. 3b).

Next, we performed in vivo $Ca^{2+}$ imaging to examine somatic activity of sensory neurons expressing the genetically encoded $Ca^{2+}$ indicator GCaMP6s in the DRG of awake, vertebrae-restrained mice (Fig. 3c–g). The first imaging session was performed > 4 days after window implantation to minimize the potential effects of surgery-related inflammation. In quiet awake mice, we found that $14.1 \pm 1.4\%$ of DRG sensory neurons (343 neurons from seven mice) exhibited spontaneous $Ca^{2+}$ transients with a duration of $5.8 \pm 0.4$ s (Fig. 3c, d). In mice anesthetized with ketamine and xylazine, the fraction of spontaneous active cells dropped to $4.3 \pm 0.5\%$ (299 neurons from six mice) (Fig. 3c, d), and the average integrated activity dropped > 90% compared to pre-anesthesia awake state (Fig. 3c, e and Supplementary

Movie 1). In awake mice, although individual neurons did not always exhibit $Ca^{2+}$ transients across all the imaging sessions (Fig. 3f and Supplementary Movie 2), the percentages of active neurons remained rather constant among sessions (1 day: $13.8 \pm 1.2\%$, 2 days: $14.0 \pm 2.1\%$, 7 days: $15.9 \pm 2.1\%$, 14 days: $15.6 \pm 1.1\%$, 21 days: $13.7 \pm 1.5\%$; $n = 109$ neurons from three mice). Moreover, total integrated $Ca^{2+}$ activity in the somas of active neurons remained comparable across imaging sessions over 3 weeks (Fig. 3g).

**Imaging noxious stimuli-evoked activity in DRG neurons.** To further investigate DRG activity in awake mice, we measured noxious stimuli-evoked sensory neuron responses. Here, we used in vivo two-photon $Ca^{2+}$ imaging to examine the activity of sensory neurons in L4 DRG in response to inflammatory pain induced by formalin[34]. In this experiment, *Thy1*-GCaMP6s-expressing mice received an ipsilateral plantar saline injection on day 1 and a formalin injection on day 2. The same group of neurons were imaged before, during, and after each injection (Fig. 4a, b). We found that saline injection elicited $Ca^{2+}$ transients in 25.1% (83/331) neurons (Fig. 4a–c and Supplementary Movie 3). The responsive cells showed a moderate increase in activity immediately upon saline injection ($P = 0.031$ in first 5 min), which gradually returned to the pre-injection baseline within 10 min ($P = 0.242$) (Fig. 4d). Compared to saline injection, formalin injection elicited a much higher increase in the activity of DRG neurons (Fig. 4a–c and Supplementary Movie 4). A robust elevation in $Ca^{2+}$ transients was observed in 53.5% (177/331) neurons, and the majority of formalin-activated neurons

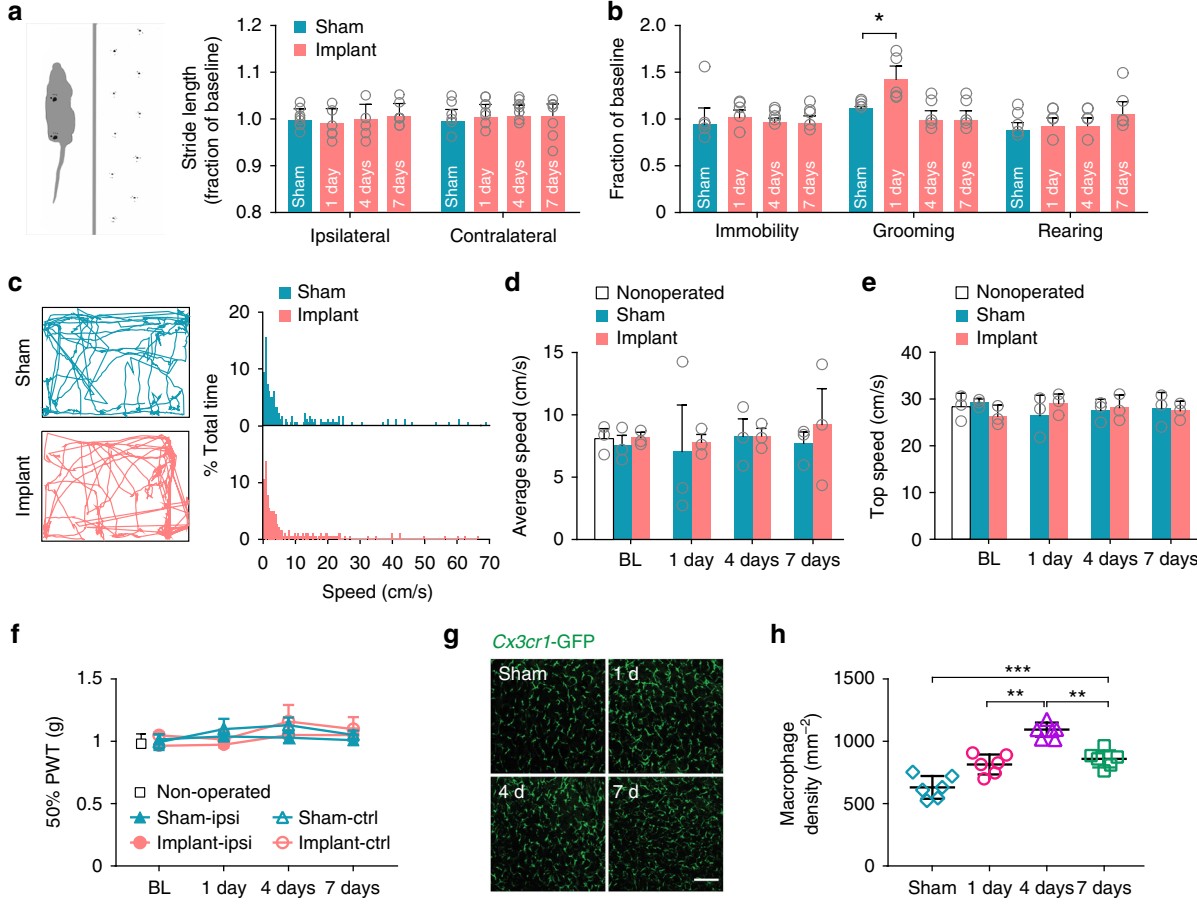

**Fig. 2** DRG window implantation has no effects on motor and sensory functions. **a** Gait analysis. Left: Illustration showing the gait pattern of a mouse running down an enclosed track. Right: Stride lengths of sham-operated mice, and mice with vertebral window implantation. Data were expressed as a fraction of the preoperative within-subject baseline ($n = 8$ mice per group). **b** The cumulative time mice spent rearing, grooming, and immobile during a 5-min open field test. Data were expressed as a fraction of the preoperative within-subject baseline ($n = 5$ mice per group). **c–e** Locomotion behavior. **c** Left: representative trajectories of an implanted and a sham-operated mouse during a 5-min open field test 4 days after surgery. Right: Speed distribution of implanted and sham-operated mice ($n = 3$ mice per group). **d** and **e** Average and top speed in the locomotor test. Top speed was calculated from the speed histogram for each mouse as the mean of the speeds above 75 percentile ($n = 3$ mice per group). **f** 50% mechanical paw withdrawal thresholds ($n = 6$ mice per group). BL: baseline, measured the day before surgery. ipsi: ipsilateral to the exposed DRG; ctrl: contralateral to the exposed DRG. **g** Representative images of *Cx3cr1*-GFP positive macrophages in L4 DRG. Scale bar, 100 μm. **h** Macrophage densities in the DRG of implanted and sham-operated mice ($n = 6$ mice per group; 1 day: 814.1 ± 32.7, 4 days: 1092.8 ± 20.7, 7 days: 859.7 ± 28.0, sham: 630.7 ± 37.4; 1 day vs. 4 days, $P = 0.002$, 7 days vs. 4 days, $P = 0.003$; 7 days vs. sham, $P = 0.0006$). Data are presented as means ± s.e.m. *$P < 0.05$, **$P < 0.01$, ***$P < 0.001$, (RM) one-way ANOVA followed by Dunnett's multiple comparisons test was used to compare time points within implant groups, and unpaired *t*-test was used to compare implant with sham in **a**, **b**, **h**, RM two-way ANOVA followed by Tukey's multiple comparisons test was used in **d–f**. The source data underlying **a–f** and **h** are provided as a Source Data file.

(138/177) displayed a prolonged Ca²⁺ elevation after the initial rise. The increase in total integrated Ca²⁺ activity was prominent upon formalin injection ($P < 0.001$ in first 5 min) and persisted longer than 10 min ($P < 0.01$) (Fig. 4d). Together, these in vivo Ca²⁺ recordings show that formalin induces a potent elevation in sensory neuron activity in affected L4 DRG.

In *Thy1*-GCaMP6s mice, GCaMP expression was observed in sensory neurons in a variety of sizes (Fig. 4e, f). We found that the majority of neurons showing high Ca²⁺ responses (>10 fold higher than baseline) after formalin injection were small-sized and medium-sized neurons. Few large-sized neurons showed increased activity after injection (Fig. 4g). Previous studies have shown that small-sized and medium-sized neurons in the DRG mediate nociceptive behavioral responses to painful stimuli, whereas large-sized neurons are related to the transduction of proprioception and low-threshold innocuous tactile

stimulation[35,36]. Our findings are consistent with these studies, suggesting that the elevated neuronal responses in formalin-injected mice are related to formalin-induced nociceptor activation in small-sized and medium-sized neurons.

**DRG neuronal activity correlates with phasic pain behavior.** It has been reported that formalin injection elicits two nociceptive phases: an early phase that occurs immediately upon formalin injection, and a late phase that arises around 30 min after injection[37]. To investigate formalin-evoked sensory neuronal activity further, we performed two-photon Ca²⁺ imaging in the somas of DRG neurons over a period of 60 min (Fig. 5a, b). In addition to the initial episode of Ca²⁺ elevation immediately after formalin injection, we observed three peaks of Ca²⁺ elevation that occurred 16–20, 24–28, and 40–44 min after formalin injection,

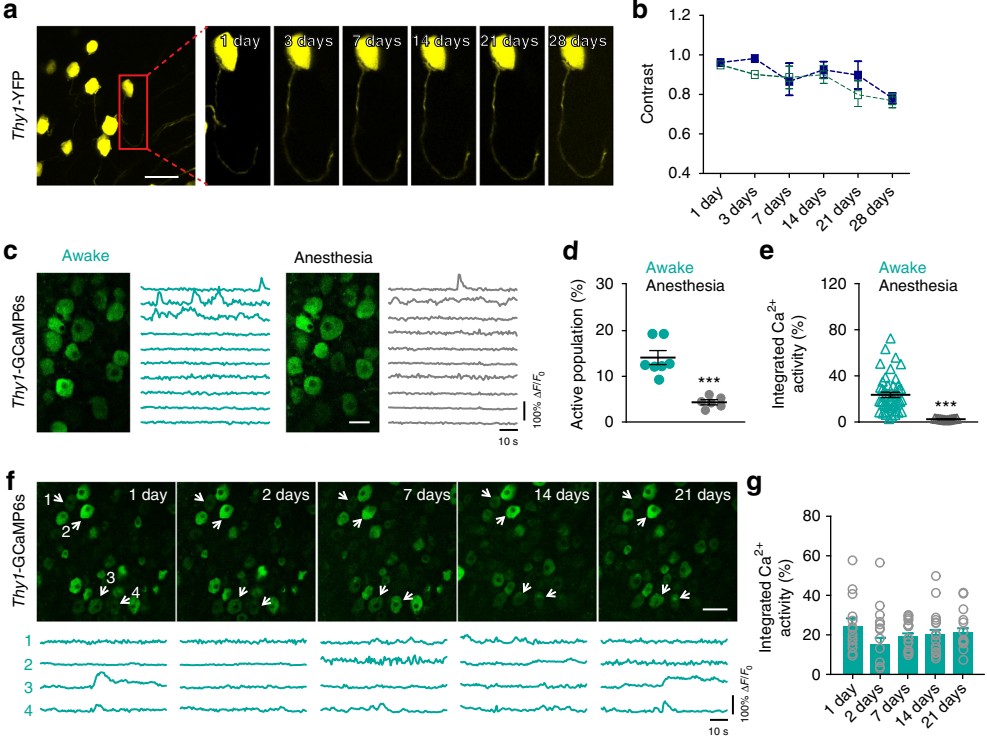

**Fig. 3** Chronic two-photon imaging of DRG sensory neurons over time. **a** Representative two-photon images of DRG sensory neurons expressing YFP. Scale bar, 50 μm. **b** Image contrast is depicted as a function of time after surgery ($n = 34$ neurons, two mice; each curve represents data from an individual animal). **c** Representative two-photon images and fluorescence traces of DRG sensory neurons expressing GCaMP6s in mice under awake (green) and anesthesia (gray) states. Scale bar, 30 μm. **d** Percentages of DRG sensory neurons exhibiting $Ca^{2+}$ transients in resting awake and anesthetized states (awake: $14.12 \pm 1.35\%$, $n = 343$ neurons from seven mice; anesthesia: $4.33 \pm 0.50\%$, $n = 299$ neurons from six mice; $t_{11} = 6.35$, $P < 0.001$, Student's $t$-test). **e** Integrated $Ca^{2+}$ activity of spontaneous active neurons in awake ($23.37 \pm 2.29$, $n = 48$ neurons from seven mice) and anesthetized mice ($2.06 \pm 0.13$, $n = 13$ neurons from six mice; $t_{59} = 4.82$, $P < 0.001$, Student's $t$-test). **f** Representative two-photon images and fluorescence traces of DRG sensory neurons expressing GCaMP6s over 21 days. Scale bar, 50 μm. **g** Integrated $Ca^{2+}$ activity of active neurons over time (1 day: $22.18 \pm 3.48$, 2 days: $14.11 \pm 3.46$, 7 days: $20.19 \pm 1.69$, 14 days: $20.98 \pm 2.65$, 21 days: $19.46 \pm 2.75$; $n = 109$ neurons from three mice; $F_{2.8,45.4} = 1.629$, one-way RM ANOVA followed by Dunnett's multiple comparisons test). Error bars, s.e.m. ***$P < 0.001$. The source data underlying **b**, **d**, **e**, and **g** are provided as a Source Data file

respectively (Fig. 5c, d). Notably, this elevated sensory neuronal activity occurred in parallel with the flinching behavior that were observed within 1 h after formalin injection (Fig. 5e, f). There was a significant correlation between sensory neuronal activity in the DRG and the animals' flinching behavior (Pearson $r = 0.7137$, $P < 0.001$) (Fig. 5g).

Previous studies have shown that formalin-evoked pain can be alleviated by cooling analgesia through TRPM8-dependent mechanisms[38,39]. To further investigate the relationship between sensory neuron activity and pain behavior, we topically applied L-menthol, an agonist of TRPM8 receptors[40], to formalin-injected hind paw (Fig. 5a). We found that L-menthol had no effect on the initial episode of sensory neuronal hyperactivity (Fig. 5b–d) and painful flinches (Fig. 5e, f), but reduced both parameters in the latter episodes (16–20 min: $P < 0.001$; 24–28 min: $P = 0.6435$; 40–44 min: $P < 0.001$; late phase: $P < 0.001$) (Fig. 5b–f). Consistent with the effect of TRPM8 receptor activation on formalin-evoked pain[34,41], these results indicate that the activation of TRPM8 receptors primarily affects late, but not early, phases of DRG neuronal activity after formalin injection.

**Long-term imaging reveals persistent neuronal hyperactivity.** To study DRG neuronal activity during the development of chronic pain, we longitudinally imaged the same population of sensory neurons in awake mice for >5 weeks after formalin injection (Fig. 6a). We found that $Ca^{2+}$ activity in the somas of sensory neurons continually increased within the first week after

formalin injection (Fig. 6b). Four to six days after formalin injection, $Ca^{2+}$ activity in DRG neurons were about ten fold higher than their pre-injection baseline (Fig. 6b). Nine to 18 days after formalin injection, these neurons continued to exhibit higher levels of $Ca^{2+}$ activity as compared to the baseline, although to a lesser degree (3–5 fold). Twenty-seven days after formalin injection, the activity of sensory neurons in the DRG returned to the pre-injection level (Fig. 6b).

The above results reveal persistent hyperactivity of sensory ganglion neurons over months after formalin injection, which may contribute to the development of chronic pain. Previous studies have shown that formalin-induced pain may persist for days after the initial injection[42,43]. Because formalin-induced spontaneous licking or flinching behavior was no longer present after the day of injection, we employed a two-compartment conditional place preference (CPP) test[44] to assess the presence or absence of ongoing pain[45]. In this test, mice were allowed free access to both compartments of CPP chamber during the preconditioning phase. Conditioning phase was performed 4–6 days after a single plantar formalin or saline injection, during which one compartment was paired with intrathecal vehicle injection and the other was paired with an analgesic drug lidocaine. During the test phase, mice were allowed free access to both compartments without injections (Fig. 6c). While saline-injected control mice showed no preference for either compartment during CPP test, formalin-injected mice preferred to stay in the compartment associated with lidocaine treatment (vehicle-

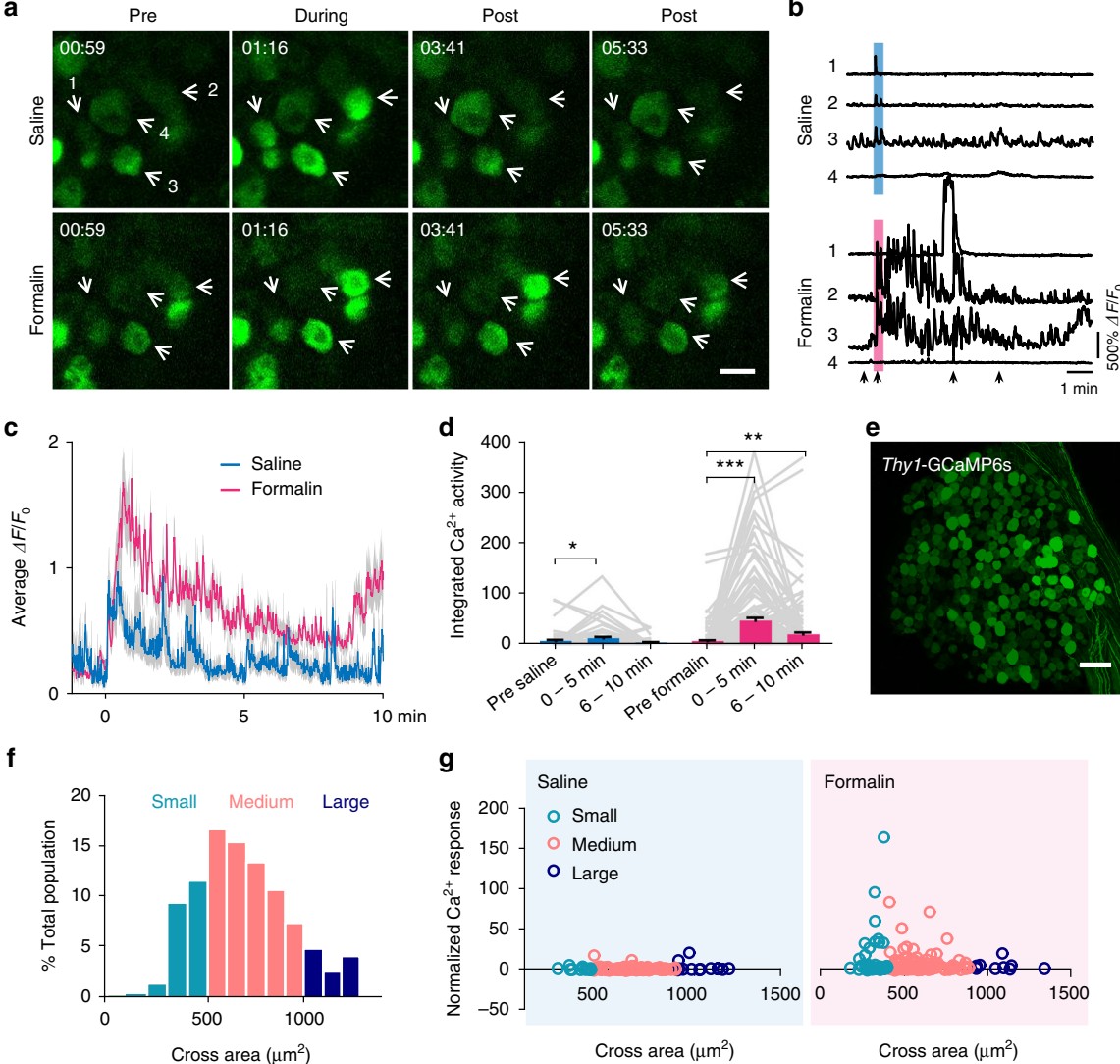

**Fig. 4** Plantar formalin injection increases sensory neuron activity in L4 DRG. **a** Representative images of GCaMP6s-expressing DRG neurons before, during, and at different time points after plantar saline or formalin injection. The time relative to the start of the imaging was marked in **a** and indicated by arrows in **b**. Scale bar, 20 μm. **b** Ca²⁺ traces of indicated neurons in **a**. Blue and pink box indicates the 20-s saline and formalin injection period, respectively. **c** Population average $\Delta F/F_0$ of all sensory neurons following saline ($n = 83$ neurons from three mice) or formalin ($n = 177$ neurons from six mice, of which 83 neurons from three mice were shared with saline group) injection. Blue and pink line denotes the mean value, and gray envelope denotes the s.e.m. **d** Integrated Ca²⁺ activity of sensory neurons before and after saline (pre-saline: 6.13 ± 2.06, 0–5 min: 11.45 ± 2.71, 6–10 min: 2.94 ± 0.90; $n = 83$ neurons from three mice; $F_{1.9, 127.3} = 6.57$, RM one-way ANOVA) or formalin (pre-formalin: 5.88 ± 1.71, 0–5 min: 46.31 ± 5.45, 6–10 min: 19.07 ± 3.82; $n = 177$ neurons from six mice; $F_{1.8, 288.5} = 40.2$, RM one-way ANOVA) injection. **e** A representative image showing GCaMP6s-expressing sensory neurons in L4 DRG. Scale bar, 100 μm. **f** Somatic size distribution of GCaMP6s-expressing neurons in L4 DRG ($n = 3542$ neurons from four mice). **g** Saline and formalin-induced Ca²⁺ responses were normalized by baseline activity and depicted as a function of somatic size. Error bars, s.e.m. *$P < 0.05$; **$P < 0.01$; ***$P < 0.001$; RM one-way ANOVA followed by Dunnett's multiple comparisons test in **d**. The source data underlying **c**, **d**, **f**, and **g** are provided as a Source Data file

paired vs. lidocaine-paired side: 202.3 ± 13.0 s vs. 392.3 ± 13.6 s; $P < 0.001$) (Fig. 6d, e). These results indicate the presence of ongoing pain 6 days after formalin injection. They also suggest that the persistent hyperactivity of DRG neurons after formalin injection may contribute to the development of chronic pain.

## Discussion

In this study, we developed a new method allowing, for the first time, long-term imaging of DRG in awake behaving mice. We showed that the structure and activity of sensory neurons in the DRG of awake mice can be reliably imaged over extended periods of time through an implanted vertebral window, without compromising the animals' motor and sensory functions. The activity of DRG neurons is low under the quiet resting condition, but increases robustly in mice with formalin-evoked inflammatory pain. This DRG hyperactivity persists over several weeks and is in parallel with the development of chronic pain.

Dorsal root ganglia are located in the neural foramen of vertebrae and contain cell bodies of peripheral sensory afferents[1]. Probing the structure and function of DRG in live animals using two-photon imaging or electrophysiological recording approaches has been challenging due to the location of DRG and the animals' body movements. Consequently, the majority of research on DRG has been conducted in vitro using cultured DRG cell or slice preparations[7–12]. The first in vivo study of DRG was reported in 2016, in which Ca²⁺ imaging was performed to examine the activity of sensory neurons expressing GCaMP in mice under

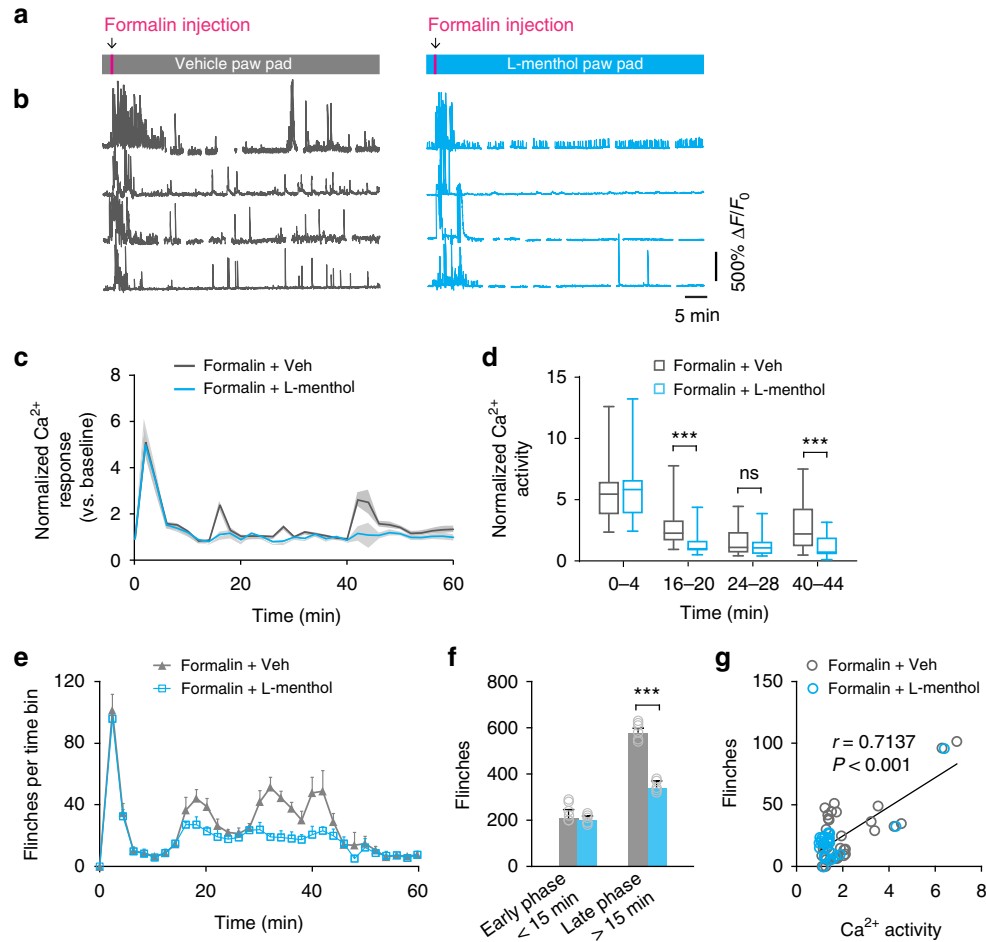

**Fig. 5** Elevated $Ca^{2+}$ responses in DRG sensory neurons are associated with ongoing pain. **a** Experimental design. Pink box indicates the 20-s formalin injection period. Gray and blue boxes indicate vehicle and L-menthol pad application on the plantar surface of the injected hind paw. **b** Representative fluorescence traces of DRG sensory neurons expressing GCaMP6s in control or L-menthol-treated mice over 60 min. **c** Population average response of DRG neurons after formalin injection in mice treated with L-menthol ($n = 97$ neurons from four mice) or vehicle ($n = 73$ neurons from three mice). Lines denote the mean and gray envelopes denote s.e.m. **d** Normalized neuronal $Ca^{2+}$ activity over 4-min recording (L-menthol vs. vehicle, 0–4 min: $5.66 \pm 0.28$ vs. $5.50 \pm 0.24$, $P = 0.947$; 16–20 min: $1.30 \pm 0.09$ vs. $2.56 \pm 0.13$, $P < 0.001$; 24–28 min: $1.27 \pm 0.10$ vs. $1.57 \pm 0.11$, $P = 0.6435$; 40–44 min: $1.22 \pm 0.10$ vs. $2.94 \pm 0.21$, $P < 0.001$; two-way ANOVA). Ends of the whiskers represent the minimum and maximum, the ends of the box are the upper and lower quartiles, and the band inside the box marks the median. **e** Number of flinches in the injected hind paw is depicted as a function of time after formalin injection ($n = 6$ mice per group). **f** Effects of L-menthol on formalin-induced flinching behavior during early (0–15 min: L-menthol vs. vehicle, $204 \pm 7.44$ vs. $220.8 \pm 9.67$, $P = 0.3715$) and late phases (16–60 min: L-menthol vs. vehicle, $359.1 \pm 8.42$ vs. $589.5 \pm 10.67$, $P < 0.001$) ($n = 6$ mice per group; two-way ANOVA). **g** DRG neuronal $Ca^{2+}$ activity after formalin injection correlates with the animals' flinching behavior (Pearson $r = 0.7137$, $P < 0.001$; $n = 267$ neurons from seven mice). Error bars, s.e.m. ***$P < 0.001$, two-way ANOVA followed by Tukey's multiple comparisons test was also used to compare L-menthol vs. vehicle at individual time points in **c**–**e**. The source data underlying **c**–**g** are provided as a Source Data file

general anesthesia[11,12]. In this study, the time window for image collection is restricted to a few hours after surgical preparation, which excludes long-term studies in the DRG. Additionally, because the animal has to be under general anesthesia during imaging, the neuronal responses evoked by sensory stimuli are greatly reduced and do not represent those occurring in the awake state (Fig. 3)[46].

The method developed in our study allows monitoring sensory stimuli-evoked responses in the DRG of awake behaving mice over minutes to weeks. A critical component of our DRG window is the custom-designed vertebral mount that creates intervertebral fusion by clamping adjacent vertebral bodies with two L-shaped plates, minimizing spinal movement during imaging. Motion-related artifacts could be further reduced by placing the animal in a custom-made cylinder during image collection if necessary. Using this method, in vivo imaging in the DRG can be readily

achieved without interfering the animal's breathing, heartbeat and limb motion. We validated that DRG window implantation had no significant effects on the animals' motor or sensory functions. There was a transient increase in macrophage density in the DRG within 4 days after window implantation, which started to subside afterwards. This transient increase in DRG macrophages is consistent with previous studies using cranial[47] or spinal cord windows[32] but far less than nerve injury[48] and peripheral inflammation[49] induced increase in immune cells.

DRG neurons detect peripheral stimuli and transmit the signal to the dorsal horn of the spinal cord, and then through thalamus to numerous brain regions[50,51], producing a diverse set of sensations, emotions, and actions. As critical components of the somatosensory system, abnormalities of sensory neurons have been documented in a variety of sensory disorders[52–55], but little is known about their activity under normal physiological states.

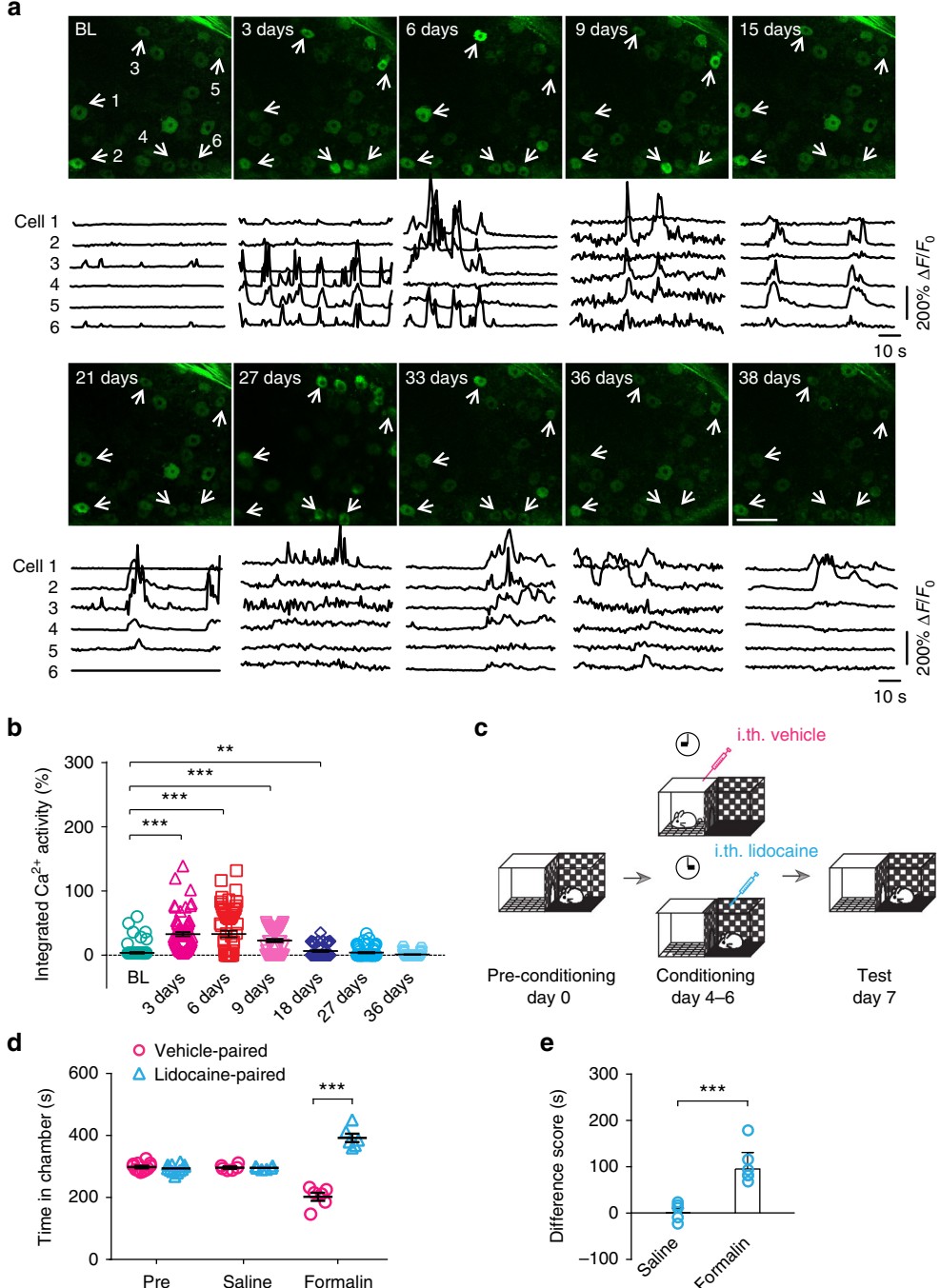

**Fig. 6** Plantar formalin injection induces long-lasting elevation of neuronal Ca$^{2+}$ activity. **a** Representative two-photon images and fluorescence traces of DRG sensory neurons expressing GCaMP6s, before and after formalin injection. Scale bar, 100 μm. **b** Integrated Ca$^{2+}$ activity of DRG neurons over time after formalin injection (BL: 4.43 ± 1.55, 3 days: 46.81 ± 4.24, 6 days: 43.57 ± 5.70, 9 days: 22.13 ± 2.04, 18 days: 11.95 ± 2.01, 27 days: 7.16 ± 1.60, 36 days: 1.94 ± 0.61; $P < 0.001$ (3 days vs. BL), $P < 0.001$ (6 days vs. BL), $P < 0.001$ (9 days vs. BL), $P = 0.009$ (18 days vs. BL), $P = 0.5183$ (27 days vs. BL), $P = 0.2619$ (36 days vs. BL); $n = 177$ cells from seven mice; $F_{1.8, 88.7} = 63.68$, RM one-way ANOVA). **c** Schematic of CPP paradigm for spontaneous pain assessment. Pre-test was carried out on day 0. Plantar saline/formalin injection was performed on day 1. Alternative conditioning took place from day 4 to 6. On day 7, a 10-min test was performed. **d** Time spent in vehicle-paired and lidocaine-paired chamber before (pre) and 6 days after saline or formalin injection (saline-injected mice: 296.17 ± 3.72 s (vehicle-paired side) vs. 295.50 ± 1.94 s (lidocaine-paired side); formalin-injected mice, 202.33 ± 13.03 s (vehicle-paired side) vs. 392.33 ± 13.63 s (lidocaine-paired side); $P < 0.001$; $n = 6$ mice per group; RM two-way ANOVA). **e** Difference score was calculated as the difference in time spent in lidocaine-paired chamber between post-test and pre-test (saline-injected group: 2.33 ± 8.24, formalin-injected group: 98.5 ± 15.4; $n = 6$ mice per group; $t_{10} = 5.50$, $P = 0.0003$, unpaired $t$-test). Error bars, s.e.m. ***$P < 0.001$. ANOVA followed by Dunnett's multiple comparisons test in **b**, **d**. The source data underlying **b**, **d**, and **e** are provided as a Source Data file

We monitored the activity of DRG by in vivo $Ca^{2+}$ imaging of sensory neurons expressing GCaMP6s, which produces large fluorescence transients (~20% $\Delta F/F$) in response to single action potential[56]. Our results show that ~14.1% DRG sensory neurons display spontaneous activity in awake mice at rest, while this active population drops to ~4.3% in anesthetized mice (Fig. 3c, d). Consistently, previous studies with electrophysiological recordings have shown that under normal physiological conditions, only ~3% DRG neurons exhibit spontaneous activity in vitro[57] and <4% in vivo under deep anesthesia[58]. A recent in vivo GCaMP-imaging study has also reported that ~5% DRG neurons show significant spontaneous calcium fluctuations under deep anesthesia[59]. The higher percentages of spontaneously active neurons we observed in awake animals could be attributed to proprioception and viscera sensation signals that are partially or completely blocked under deep anesthesia.

Very little is known about the activity of DRG sensory neurons during the development and resolution of chronic pain. The implanted vertebral window allows us to follow the activity of individual cells in the DRG of awake animals over weeks and correlate them with the animals' pain-related behavior. Formalin injected into the hind paws of freely moving rodents evokes two temporally distinct phases of spontaneous pain behavior manifested as licking and flinching of the injected limb[60,61]. By imaging DRG neuronal activity, we uncovered distinct phases of sensory neuronal excitation immediately following formalin injection. Moreover, the episodes of sensory neuronal firing positively correlated with the animals' flinching behavior. Previous electrophysiological studies have shown that formalin injection produces a distinct biphasic excitatory response in dorsal horn neurons[41]. Since these spinal cord dorsal lamina I and II neurons relay nociceptive signals from the DRG to the supraspinal area[50], the biphasic response in the dorsal horn is likely triggered by the phasic activity of DRG neurons. Consistent with previous reports that formalin-induced pain can be alleviated by cooling analgesia through the involvement of TRPM8[38], we show that topical application of L-menthol, a TRPM8 agonist, to the formalin-injected paw significantly reduces the activity of DRG sensory neurons, as well as relieves the animals' flinching behavior. In addition, a persistent elevation of sensory neuronal activity occurs in the DRG up to 21 days after formalin injection, which correlates with the increased preference of mice to analgesia-paired chamber during CPP test, indicating the presence of ongoing pain. Together, these findings suggest that the upsurge of DRG sensory neuron activity may serve as a cellular indicator for the presence of ongoing pain.

In summary, the in vivo two-photon imaging technique we developed here allows us to monitor the activity of individual cells in the DRG of awake mice over extended periods of time. In combination with transgenic targeting or viral gene transfer of fluorescent indicators, this method provides a direct window to investigate sensory neuron structure and function under both physiological and pathological states. We envision that DRG imaging in awake animals will greatly facilitate the studies of sensory ganglionopathy and pain prognosis, as well as the development of effective therapies in the future.

## Methods

**Experimental animals**. Transgenic mice expressing GCaMP6s in a subset of DRG sensory neurons, Thy1-GCaMP6 slow founder line 3, were generated at New York University School of Medicine. Thy1-YFP-H mice (stock no. 003782) expressing YFP in DRG sensory neurons and Cx3cr1-GFP mice (stock no. 005582) expressing GFP in DRG macrophages were purchased from the Jackson Laboratory.

Two-month to 3-month-old animals of both sexes were used for all the experiments. Mice were group-housed in temperature-controlled rooms on a 12-h light–dark cycle. After vertebral window implantation, mice were housed individually to minimize the risk of window damage. All animal procedures were performed in accordance with protocols approved by the Institutional Animal Care and Use Committee (IACUC) of New York University and Columbia University as consistent with National Institutes of Health (NIH) Guidelines for the Care and Use of Laboratory Animals.

**Surgical procedure for vertebral window implantation**. Mice were deeply anesthetized with an intraperitoneal injection of 100 mg kg$^{-1}$ ketamine and 15 mg kg$^{-1}$ xylazine. Surgical plane of anesthesia was verified by a lack of toe pinch reflex. Ocry-gel was applied to both eyes to maintain eye moisture. After shaving the fur at the surgical site and sterilizing the skin with 70% ethanol and 10% povidone-iodine, a small incision was made in the dorsal skin at the L3–L5 level of the spine and the skin was held back with retractors. Under a stereomicroscope, muscles and ligaments attached to the lateral aspects of three vertebrae were detached using surgical scissors. The vertebral mount consisted of two L-shaped plates. The prong arms of plates with anti-skid grooves were inserted to the lateral aspects of L4 and L5 vertebra. Two mounting plates were aligned and loosely screwed (stop fastening the screw once resistance is encountered) to each other (Supplementary Fig. 3). The mouse was then placed lying on its right side with the left lateral aspect of L4 vertebra facing upright. The arm of the long plate was plugged and secured to a heavy metal base, and the screw in the mounting plates was further tightened to fix the vertebral mount in place.

To prepare a vertebral window, the muscle tissue near L4 DRG was held back with a thin aluminum sheet and the bone was scraped clean, including the distal half of L4, surrounding region of L4 neural foramen and proximal half of L5. After removing connective tissues covering the surface of neural foramen, L4 DRG was exposed. Using a high-speed micro-drill, articular processes around DRG were gently trimmed to the same level of the DRG surface. Sterile cotton applicators were used to control bleeding and the spinal cord was irrigated with sterile saline. After hemostasis, DRG was covered with a thin layer of Kwik-Sil® silicone elastomer and sealed with a 2-mm diameter cover glass. Lumbar 4 DRG is ~1.5 mm in diameter in the lateral direction. With a 2-mm glass window, the entire DRG can be accessed for imaging. When placing the cover glass, it was critical to ensure no blood or excessive saline on top of the DRG, which may impede the adhesion of the elastomer, rendering it susceptible to detach from the DRG, leading to a blurry window. The cover glass was attached to the vertebral mount by cyanoacrylate glue and dental acrylic. An aluminum ring (2 mm in diameter) edged dental acrylic, serving as the frame of DRG window. Throughout the surgical procedure and recovery, the animal's body temperature was maintained at ~37 °C.

**Locomotor and sensory tests**. Runway assays were conducted in a Plexiglass enclosure (80 cm long × 8 cm deep × 15 cm high). During the test, mice with inked paws traversed the length of runway without pausing or prompting and entered a goal box at the end of runway. Paper was placed on the floor of runway to collect footprints. Stride length was measured as the distance between the central pads of two consecutive hindlimb prints on the left or right. Runway assays were conducted in three trials per day on the day before surgery, and 1, 4, and 7 days after surgery. Footprint measurements were made from five consecutive steps in each trial. All measurements were made at the same time of day to avoid circadian variability.

To measure grooming and rearing time, mice were placed in the center of a plexiglass enclosure (46 cm × 42 cm × 30 cm) and recorded for 5 min. Video tracking analysis for rearing, grooming, and immobility was performed using ANY-maze software (Stoelting, Wood Dale, IL). Grooming was defined as any period during which the animal licked its fur or moved its forelimbs over the head. Rearing was defined as any period during which the animal lifted both of its forelimbs off the ground simultaneously.

Dixon's up and down method was employed to measure the animals' paw withdrawal threshold[62]. In short, mice were individually placed into transparent acrylic boxes (10 cm × 7 cm × 7 cm) over a mesh table and habituated for at least 30 min. A series of von Frey fibers (0.008, 0.02, 0.07, 0.16, 0.4, 0.6, 1.0, 1.4, 2.0, and 4.0 g) were then presented in consecutive ascending order. In the absence of paw withdrawal response, the next stronger stimulus was presented; in the event of paw withdrawal, the next weaker stimulus was chosen. After the response threshold was first crossed, six data points were counted, at which time the two responses straddling the threshold were retrospectively designated as the first two responses of the series of 6. 50% response threshold was calculated as: 50% g threshold = $(10^{[X_f + \kappa\delta]})/10,000$, where $X_f$ = value (in log units) of the final von Frey hair used; $\kappa$ = tabular value for the pattern of positive/negative responses[63]; and $\delta$ = mean difference (in log units) between stimuli (here, 0.2699).

**In vivo two-photon imaging**. In vivo imaging of L4 DRG in awake mice was performed > 4 days after surgery. To minimize motion artifacts during imaging, the animal was vertebrae restrained. In addition, mice were placed in a 2.9-cm-diameter transparent plastic cylinder to further attenuate motion artifacts. The cylinder had a custom-made window that exposed the DRG window for imaging, and the arm of vertebral mount for fixation to metal base. This cylinder could minimize intense struggle, but had less effect on the movements of limbs (Supplementary Fig. 4). Thereafter, the mouse together with the cylinder was mounted

onto a heavy metal base. Mice were habituated for at least three trials (10 min per trial) before imaging.

The genetically encoded $Ca^{2+}$ indicator GCaMP6 slow (GCaMP6s) was used for $Ca^{2+}$ imaging of sensory neurons in the DRG. The in vivo imaging experiments were performed using a Bruker Investigator two-photon system equipped with a DeepSee Ti:sapphire laser (Spectra Physics) tuned to 920 nm. The average laser power on the sample was ~20–30 mW. L4 DRG is 180–250 μm in depth. For $Ca^{2+}$ imaging, images were collected 30–200 μm below the DRG surface at frame rates of 1.3–1.7 Hz at a resolution of 512 × 512 pixels using a ×25 objective (NA = 1.05) immersed in artificial cerebrospinal fluid and with a 1 × digital zoom. There was no marked difference in image quality at different depths of DRG. Image acquisition was performed using Bruker PrairieView software. The imaging parameters were chosen to allow repeated imaging of the same cells without causing damage to cells and surrounding tissues. To evaluate the imaging quality of DRG window over time (Fig. 3a–b), the experimental conditions were kept exactly the same throughout all the imaging sessions to ensure proper calculation of image contrast.

**Imaging data analysis.** During quiet resting, body movements were infrequent and minimized by habituation, with the use of vertebral mount and restraining cylinder. Motion-related artifacts derived from respiration and heart beat were typically <2 μm as detected in DRG measurements. When the animal struggled during image collection, images from those segments were excluded from quantification. All imaging stacks were registered using NIH ImageJ plug-in StackReg.

In this study, the somas of DRG neurons were labeled with GCaMP6s in the cytoplasm. The somas with nuclear expression (<1%) were excluded from analysis. Regions of interests (ROIs) corresponding to visually identifiable somas were selected for quantification at the depth of 30–200 μm below the surface of DRG. The fluorescence time course in each soma was measured in NIH ImageJ by averaging all pixels within the ROI covering the soma. The $\Delta F/F_0$ was calculated as $\Delta F/F_0 = (F - F_0)/F_0 \times 100\%$, where $F_0$ is the baseline fluorescence signal averaged over a 2-s period corresponding to the lowest fluorescence signal over the recording period. Active DRG neurons were defined with the criteria that calcium transients were beyond the threshold of three times standard deviation of baseline. In order to compare neuronal activity among different cells and at different time points, we performed an integrated measurement of a cell's output activity over 2-min recording, termed total integrated calcium activity. Total integrate calcium activity is the average of $\Delta F/F$ over 2 min.

**Formalin-evoked flinching behavior.** Inflammatory pain was induced by subcutaneous formalin injection. Specifically, 10 μL 2% formalin solution was injected under the plantar surface of the left hindpaw with a 30-gauge needle. To administer formalin during imaging experiments, the infusion needle was connected to a microinjector through the flexible tubing. For cooling analgesia, a cotton pad soaked with 1% L-menthol (Sigma, dissolved in 75% ethanol) was placed on the plantar surface of the left hindpaw under thermoneutral conditions. Control mice received a cotton pad soaked with 75% ethanol.

To measure formalin-induced flinching behavior, mice were placed into a transparent observation chamber (30 cm × 30 cm × 25 cm) and habituated to the environment for 30 min. After formalin injection, mice were observed for the flinching behavior. Flinches were counted during 2-min intervals, starting from the time of formalin administration through 60 min post injection.

**Conditioned place preference (CPP) test.** The CPP test was performed in two-compartment CPP chamber (Ugo basile; Varese, Italy)[44]. The chamber (32 cm × 15 cm × 25 cm) consisted of two compartments of the same size that were connected through a removable door (4 cm wide × 6 cm high). Each compartment had different visual and textured cues (e.g., wall patterns, floor patterns, and texture). To minimize the effect of novelty and stress associated with exposure to the chamber during lidocaine CPP, mice were habituated to the chamber for 10 min for 3 consecutive days. In the pre-conditioning phase (day 0), animals were placed in CPP chamber with the door removed, and recorded for 10 min. The time spent in each compartment was recorded. The animals showing strong unconditioned preference (>400 s) were excluded. On day 1, animals were randomly assigned for plantar formalin injection or saline injection. During conditioning (day 4–6), mice were injected with vehicle (5 μL saline, i.th.) in the morning and immediately confined to the paired compartment for 30 min; about 6 h later, the mice were injected with lidocaine (0.04% in 5 μL saline, i.th.) and immediately placed in the other compartment for 30 min. Compartment assignments were counterbalanced among all the test mice. CPP test was carried out on day 7. Mice were allowed free access to both compartments of CPP chamber for 10 min. The time spent in each compartment was recorded for each animal, and the change of preference was calculated as difference (in seconds) of time spent in the drug-paired compartment between the test and preconditioning phase: difference score = test – preconditioning.

**Statistical analysis.** Summary data are presented as means ± s.e.m. No statistical methods were used to pre-determine sample sizes but our sample sizes are similar to those reported in previous publications[15]. Data distribution was assumed to be normal but this was not formally tested. No samples or animals that were

successfully imaged or measured were excluded from the analysis. The variance was similar between the groups that were statistically compared. Tests for differences between two populations were performed using paired or unpaired $t$ test. One-way or two-way (repeated measures) ANOVA followed by Dunnett's or Tukey's test was used to compare difference among various groups. Significant levels were set at $P \leq 0.05$. All statistical analyses were performed using GraphPad Prism.

**Reporting summary.** Further information on research design is available in the Nature Research Reporting Summary linked to this article.

## Data availability

The source data underlying Figs. 2a–f, h, 3b, d, e, g, 4c, d, f, g, 5c–g, 6b, d, and e are provided as a Source Data file. All relevant data is available from the corresponding author upon reasonable request.

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

## Acknowledgements
We thank members of Gan and Yang Laboratories for helpful discussion. This work was supported by National Institutes of Health grants R21NS106469 (G.Y.), R01AA027108 (G.Y. and W.-B.G.) and R01NS047325 (W.-B.G.), and National Natural Science Foundation of China grant 81520108017 (P.T.).

## Author contributions
C.C., L.S., W.-B.G., P.T. and G.Y. designed the experiments. J.Z. designed the vertebral mount, C.C. and L.S. performed the experiments and analyzed the data. C.C., L.S., J.Z., Y.Z., W.-B.G., P.T. and G.Y. contributed to data interpretation. C.C., L.S., W.-B.G. and G.Y. wrote the manuscript.

## Additional information

**Competing interests:** The authors declare no competing interests.

