## [Peer Review File · Nature Communications]

Reviewers' comments:

Reviewer #1 (Remarks to the Author):

The paper by Chen et al describes a novel approach for long-term imaging of DRG neurons. The authors developed a new technique describing the vertebrae mount and demonstrated that they are able to record from the same DRG neurons over several weeks. In a series of experiments, they use this approach to correlate pain behaviour with activity of DRG neurons. The paper is well written, the figures are of high quality and the technical aspects of the study are sound. The design of the vertebrae mount is clearly described at the supplementary data section. My only concern is related to the extent of conceptual advance provided by this work.

In Fig. 6 the authors write "In quiet awake mice". It is not clear from this if the mice are lightly anesthetised or not.

Reviewer #2 (Remarks to the Author):

This is an interesting manuscript, which reports a new method to implant a vertebral glass window to allow chronic two photon imaging of calcium activities in dorsal root ganglia (DRG) neurons in awake mice for the first time. The clever design of an intervertebral fusion mount can reduce spinal movement required for stable subcellular resolution imaging but do not interfere the animal's motor and sensory functions, which was confirmed by animals' gait patterns and paw withdrawal threshold test paradigms. Using this new technique, authors studied the correlation between neural activities in DRG and plantar formalin injection-induced inflammatory pain behavior. The experiments were well designed and results were summarized statistically. I recommend publication of this manuscript on Nature Communications if following concerns can be properly addressed:

(1) Since the new technique is one of major novelties of the manuscript, more details of the technique should be made available to general readers. For example, what's the 'circled wire' in Figure 1d. What's the design purpose of this structure? One possible worry of this surgery is the locking of two L-shaped plate by screw. In order to ensure the stability of this attachment, a certain amount of torque has to be applied when fastening the screw, which might cause injury to the spinal cord. Would this a concern or not? More description and discussion about details like this would be very helpful to readers.

(2) In evaluation of effects of the window implantation on locomotion and sensory functions, the control group is sham-operated mice underwent skin incision and muscle/ligament detachment. Is there any reason for not using untreated mouse as control? The statement that implantation has no effect on motor and sensory functions seems to imply that the comparison is done using untreated mice as control.

(3) The conditional place preference (CPP) test design reported in this study has some difference from that in reference 44. For example, there is a 3rd box representing neutral preference in original design in reference 44 and is neglected in this experiment. It may affect the calculation of screening threshold in excluding pre-existed bias. It may also affect the test results as the mouse may prefer the central box. In reference 44, one trail conditioning is chosen and showed to be enough to obtain statistical significance. Would this the case in your study?

(4) The thy1-gcamp6s mice were employed in this work. I wonder whether the virus injection protocol, which can potentially be more flexible, is compatible with current technique. If not, would you discuss possible problems and envision possible solutions?

(5) Several questions regarding to the imaging:

a) How deep can you image into DRG? Can you cover the whole depth? The imaging depth is chosen in the range of 30-200um in this work, would you expect any difference at different depth?

- b) How wide can you cover in the lateral direction? Is there any technical limiting factor?
- c) How long can this window work properly? What's the limiting factor?
- d) Using imaging contrast to evaluate the imaging quality is appropriate, but calculation of I_{\min} can be critical. If the I_{\min} refers to background signal level, both background fluorescence signal and detector read noise can contribute to it. So the experimental conditions has to be kept the exactly the same throughout all trails.

Kai Wang

Reviewer #1 (Remarks to the Author):

The paper by Chen et al describes a novel approach for long-term imaging of DRG neurons. The authors developed a new technique describing the vertebrae mount and demonstrated that they are able to record from the same DRG neurons over several weeks. In a series of experiments, they use this approach to correlate pain behaviour with activity of DRG neurons. The paper is well written, the figures are of high quality and the technical aspects of the study are sound. The design of the vertebrae mount is clearly described at the supplementary data section. My only concern is related to the extent of conceptual advance provided by this work.

We thank the reviewer for the compliment on the novelty of our work. In this study, we developed a new method allowing, for the first time, long-term imaging of DRG in awake behaving mice. Using this awake animal DRG imaging approach, we showed *in vivo* that the activity upsurge in DRG sensory neurons correlated with the animals' phasic pain behavior, and DRG neuronal hyperactivity persisted in mice with chronic pain. We hope that the reviewer would agree with us that the method developed in this study opens a new avenue for *in vivo* studies of sensory ganglia-related disorders and neuropathies.

In Fig. 6 the authors write "In quiet awake mice". It is not clear from this if the mice are lightly anesthetized or not.

Experiments presented in Fig. 6 were performed in awake mice, without the use of anesthesia. We have now clarified this point in the manuscript (line 15, page 8).

Reviewer #2 (Remarks to the Author):

This is an interesting manuscript, which reports a new method to implant a vertebral glass window to allow chronic two photon imaging of calcium activities in dorsal root ganglia (DRG) neurons in awake mice for the first time. The clever design of an intervertebral fusion mount can reduce spinal movement required for stable subcellular resolution imaging but do not interfere the animal's motor and sensory functions, which was confirmed by animals' gait patterns and paw withdrawal threshold test paradigms. Using this new technique, authors studied the correlation between neural activities in DRG and plantar formalin injection-induced inflammatory pain behavior. The experiments were well designed and results were summarized statistically. I recommend publication of this manuscript on Nature Communications if following concerns can be properly addressed:

(1) Since the new technique is one of major novelties of the manuscript, more details of the technique should be made available to general readers. For example, what's the 'circled wire' in Figure 1d. What's the design purpose of this structure? One possible worry of this surgery is the locking of two L-shaped plate by screw. In order to ensure the stability of this attachment, a certain amount of torque has to be applied when fastening the screw, which might cause injury to the spinal cord. Would this a concern or not? More description and discussion about details like this would be very helpful to readers.

We thank the reviewer for encouraging and constructive comments on our work. We have now provided the detailed description of the ‘circled wire’ and its design purpose in the Method section (lines 12-14, page 13). Specifically, the ‘circled wire’ is an aluminum ring with a diameter of 2 mm. The ring serves as the frame of the glass window and prevents dental cement from sliding to the center of window before the cement becomes solid.

As the reviewer pointed out correctly, the two L-shaped plates need to be mounted appropriately to ensure the stability of the window. One should not use a large amount of torque when fastening the screw in order to avoid potential injuries to the spinal cord. This can be achieved through the following steps: First, align and loosely lock L-shaped plates to the spinal column and stop fastening the screw once resistance is encountered. Next, place the mouse on a heavy metal base with the spinal column and L-shaped plates aligned to the edge of metal base. After the long arm of the L-shaped plate is fixed to the metal base, fasten the screw further to stabilize the vertebral mount. This procedure minimizes torque applied to the spine during implantation and avoids the potential injury to the spinal cord. We have now included these details in the Method section (lines 24-30, page 12). We thank the reviewer for raising this important issue.

(2) In evaluation of effects of the window implantation on locomotion and sensory functions, the control group is sham-operated mice underwent skin incision and muscle/ligament detachment. Is there any reason for not using untreated mouse as control? The statement that implantation has no effect on motor and sensory functions seems to imply that the comparison is done using untreated mice as control.

We have now added new data from the untreated control group in both locomotion and sensory function tests (Fig. 2c-f). Consistent with the observation in gait analysis (Fig. 2a-b), we found no significant difference in both locomotion and sensory tests between un-operated mice, sham-operated mice and mice with implantation. These new data strengthen our previous statement that DRG window implantation has no obvious effect on the animals’ motor and sensory functions. We thank the reviewer for the suggestion.

(3) The conditional place preference (CPP) test design reported in this study has some difference from that in reference 44. For example, there is a 3rd box representing neutral preference in original design in reference 44 and is neglected in this experiment. It may affect the calculation of screening threshold in excluding pre-existed bias. It may also affect the test results as the mouse may prefer the central box. In reference 44, one trail conditioning is chosen and showed to be enough to obtain statistical significance. Would this be the case in your study?

We apologize for the confusion regarding the design of our CPP experiments. We have now clarified in both Result (line 26, page 8) and Method sections (lines 9-23, page 16) that the CPP test was performed in two-compartment CPP chamber (Ugo basile 42552/3; Varese, Italy) and updated the relevant reference. Comparing to three-compartment CPP, two-compartment chamber eliminates interpretational difficulties caused by test animals spending excess time in a center compartment, and has been used in many studies to measure ongoing pain and drug addiction (Asaoka et al, 2018 *Neuroscience Letter*; Zhang et al, 2016 *Neuroscience Letters*; Morales et al, 2007 *Addict Biol*). In our CPP test, we conducted three conditioning trials 4-6 days after saline or formalin injection, which consistently demonstrated the presence of ongoing pain

in formalin-treated mice. We have not performed any one-trial-conditioning CPP test similar to that in reference 45 (He *et al.*, *J Pain*, 2012). It would be interesting to test in the future whether one dose of lidocaine treatment would be sufficient to cause motivational effects in formalin-injected mice.

(4) The thy1-gcamp6s mice were employed in this work. I wonder whether the virus injection protocol, which can potentially be more flexible, is compatible with current technique. If not, would you discuss possible problems and envision possible solutions?

There are several virus delivery options for DRG infection such as spinal nerve injection, sciatic nerve injection, intrathecal injection, intraplantar injection and intraganglionic injection (Fischer *et al.*, *J Neurosci Meth*, 2011; Abdallah *et al.*, *Sci Rep-Uk*, 2018). Although we haven't performed imaging of DRG neurons expressing GCaMP6 with the viral infection method, we expect that viral gene transfer of fluorescent indicators is also compatible with the current imaging technique. We have now mentioned this point in the Discussion (line 19, page 11).

(5) Several questions regarding to the imaging:

a) How deep can you image into DRG? Can you cover the whole depth? The imaging depth is chosen in the range of 30-200um in this work, would you expect any difference at different depth?

We could image ~200 μm into DRG and cover almost the whole depth of L4 DRG in mouse, which is about 180-250 μm in depth. In time-lapse imaging experiments, we chose the imaging depth between 30 and 200 μm from the surface where we could capture more fluorescence-expressing cells. We did not observe marked differences in image quality (brightness and signal to noise ratio) at different depths of DRG. We have now added this information in the method (lines 28-31, page 14; line 1, page 15).

b) How wide can you cover in the lateral direction? Is there any technical limiting factor?

Lumbar 4 DRG is ~1.5 mm in diameter in the lateral direction. With a 2 mm-diameter glass window, the entire DRG can be accessed for imaging. We have added this information in the method (lines 6-9, page 13).

c) How long can this window work properly? What's the limiting factor?

In our hands, the DRG window can stay clear for weeks. The longest imaging duration we have achieved is 45 days. The deterioration of window quality is typically attributed to the presence of blood and inflammatory tissue between the cover glass and the surface of DRG. The critical step in window preparation is to apply a thin layer of elastomer between the DRG surface and glass window, which acts as a neuroprotective adhesive to secure the DRG and glass window in place. This layer of elastomer is applied after the local bleeding has fully stopped. Any residual blood or excessive saline can impair the adhesion of the elastomer and make it susceptible to detach from the DRG surface. Any detachment will further invite inflammatory tissue to grow in between and blur the window. We have now included this information in the Method section (lines 6-14, page 13).

d) Using imaging contrast to evaluate the imaging quality is appropriate, but calculation of I_min can be critical. If the I_min refers to background signal level, both background fluorescence signal and detector read noise can contribute to it. So the experimental conditions have to be kept the exactly the same throughout all trails.

As the reviewer pointed out correctly, the calculation of I_min (background signal level) is affected by not only background fluorescence signal but also the detector read noise. Therefore, we have kept the experimental conditions exactly the same throughout all the imaging sessions. We have added this information in the method (lines 3-6, page 15)

References

Abdallah, K., Nadeau, F., Bergeron, F., Blouin, S., Blais, V., Bradbury, K.M., Lavoie, C.L., Parent, J.L., Gendron, L. Adeno-associated virus 2/9 delivery of Cre recombinase in mouse primary afferents, *Scientific reports*, 8 (2018) 7321.

Fischer, G., Kostic, S., Nakai, H., Park, F., Sapunar, D., Yu, H.W., Hogan, Q. Direct injection into the dorsal root ganglion: Technical, behavioral, and histological observations, *J Neurosci Meth*, 199 (2011) 43-55.

He, Y., Tian, X., Hu, X., Porreca, F. & Wang, Z. J. Negative reinforcement reveals non-evoked ongoing pain in mice with tissue or nerve injury. *J Pain* 13 (2012) 598-607.

Morales, L., Perez-Garcia, C., Herradon, G., Alguacil, L.F. Place conditioning in a two- or three-conditioning compartment apparatus: a comparative study with morphine and U-50,488, *Addict Biol*, 12 (2007) 482-484.

Asaoka Y, Kato T, Ide S, Amano T, Minami M. Pregabalin induces conditioned place preference in the rat during the early, but not late, stage of neuropathic pain. *Neuroscience Letter*, 668 (2018) 133-137.

Zhang, J.B., Wang, N., Chen, B., Wang, Y.N., He, J., Cai, X.T., Zhang, H.B., Wei, S.G., Li, S.B. Blockade of Cannabinoid CB1 receptor attenuates the acquisition of morphine-induced conditioned place preference along with a downregulation of ERK, CREB phosphorylation, and BDNF expression in the nucleus accumbens and hippocampus, *Neuroscience Letters*, 630 (2016) 70-76.

REVIEWERS' COMMENTS:

Reviewer #2 (Remarks to the Author):

My concerns were addressed properly and I would recommend publication of this revised manuscript.